# LAMBDA: Assessing Few-shot Lexical Analogical Reasoning in Language Models

## Abstract

Analogical reasoning in language models is a critical yet underexplored aspect of their capability, particularly as models grow in scale and training data. This work investigates the limitations of current models in inferring latent relational structures, focusing on lexical analogies. We introduce LAMBDA, a novel dataset of 3,000 relation-hidden lexical analogies spanning synonyms, antonyms, and derivational transformations, designed for two-shot induction. Our empirical evaluation across eight models, including four open-source models from 0.1B to 17B parameters, along with four commercial models, reveals a wide performance gap, with accuracies ranging from 0.3% to 46.4%, highlighting the challenge of systematic generalization. By analyzing error patterns such as identity echo and semantic drift, we provide insights into model weaknesses. These findings suggest that large-scale pretraining alone does not guarantee strong relational reasoning abilities, offering a foundation for targeted improvements in model design. Broader implications point to the potential for refining training methodologies to enhance analogical abstraction in language models.

## 1 Introduction

Analogical reasoning is central to human cognition (Gentner, 1983; Hofstadter, 2001) and remains a frequently used test for vector-space semantics (Mikolov et al., 2013a; Levy & Goldberg, 2014). Language models have grown quickly in parameter count and training data (Brown et al., 2020; Bommasani et al., 2021; Touvron et al., 2023; Meta AI, 2025), yet their ability to infer latent relational structure rather than memorize superficial patterns is disputed (McCoy et al., 2019). This question matters because tasks such as zero-shot entity linking (Logeswaran et al., 2019) and compositional question answering (Keysers et al., 2020) rely on systematic generalization.

Classic analogy benchmarks like the Google set (Mikolov et al., 2013a) present relation labels and allow free-form decoding, letting models exploit cues or prompt tricks. Later resources such as BATS (Gladkova et al., b) and WordRep (Gao et al., 2014) widen relation coverage but still disclose the mapping, while ANALOGYKB supplies a million-scale resource for training (Yuan et al., 2024). Some studies suggest that larger scale does not guarantee compositional abstraction (Hupkes et al., 2020), motivating tasks that isolate a single skill.

We introduce LAMBDA (Lexical Analogy and Morphology Benchmark for Deep Abstraction), a corpus of 3,000 relation-hidden lexical analogies designed for two-shot induction. Items span synonyms, antonyms, and derivational transformations, created deterministically from WordNet (Miller, 1995) with strict length and overlap filters. This setup measures a model's ability to infer and apply an unseen mapping.

Baseline experiments reveal a steep performance ladder. Our weakest instruction-tuned baseline, Mistral-7B-Instruct-v0.3, answers only 4.9% of items correctly (45 synonyms, 70 antonyms, 32 derivations). For historical context we also tested GPT-2 medium, which achieves just 0.3%. At the high end, the 17B-parameter Llama-4-Maverick (henceforth also referred to as Maverick) scores 46.4%. The observed error patterns (identity echo, surface misfire, semantic drift) mirror findings in morphological generalization and

---

All data are derived from public sources (WordNet). No personally identifiable or sensitive content is included.

adversarial probing (Naik et al., 2018). With 3,000 trials, 95% binomial intervals are $\pm 1.8$ percentage points, demonstrating a separation between closely matched systems. Recent representational studies confirm that models form internal concept vectors for relations such as antonymy yet still miss correct outputs (Opiełka et al., 2025).

LAMBDA is a lightweight CC BY-SA[1] dataset that isolates analogical abstraction without confounds from extended discourse or numeric reasoning. Initial results suggest that large-scale pre-training, although vital for lexical coverage (Liu et al., 2019), does not guarantee reliable relation induction, consistent with limits observed in compositional tests for vision-language models (Kim et al., 2023). We hope to invite exploration of richer prompting (Zhou et al., 2023), targeted fine-tuning (Lu et al., 2022), and symbolic hybrids (Bogin et al., 2019) aimed at closing the gap to human-level analogy making.

**Contributions**

- **Benchmark:** We introduce LAMBDA, a 3,000-example dataset of relation-hidden lexical analogies balanced across synonymy, antonymy, and derivational morphology.

- **Reproducibility:** We release a deterministic generation pipeline along with JSONL data files so that any researcher can recreate the benchmark exactly.

- **Empirical evaluation:** We run six open-source language models spanning 0.1B–17B parameters under a strict two-shot protocol, uncovering a 150$\times$ accuracy gap between GPT-2 medium and Llama-4-Maverick-17B.

- **Error analysis:** We provide a clear taxonomy of failure modes (identity echo, surface misfire, semantic drift) that guides targeted model improvements.

- **Statistical rigor:** We compute binomial 95% confidence intervals ($\pm 1.8pp$) for every score, enabling statistically sound comparisons with forthcoming commercial systems.

## 2 Our Approach

### 2.1 Problem definition

Let $\mathcal{W}$ be the set of lowercase English lemmas in WordNet (Miller, 1995) that are alphabetic and 4–15 characters long. We study three binary relations on $\mathcal{W}$:

$$R_{\mathrm{syn}}(w) = \{w' \mid w' \text{ is a synonym of } w\}, \qquad R_{\mathrm{ant}}(w), \qquad R_{\mathrm{der}}(w),$$

corresponding to synonymy, antonymy, and derivational morphology. Given two support pairs obeying the same relation $r$,

$$(A, B),\ (C, D) \quad \text{with } B \in R_r(A),\ D \in R_r(C),$$

the model must output a single token $\hat{F}$ such that $\hat{F} \in R_r(E)$ for the query pair $(E, ?)$.

### 2.2 Dataset generation

Algorithm 1 samples items until each relation contributes exactly 1,000 valid instances, yielding 3,000 analogies. You can see Listing 1 for the full Python generation script.

---

[1] https://creativecommons.org/licenses/by-sa/4.0/

---

**Algorithm 1** Deterministic item generation

---

1: $\mathcal{D} \leftarrow \emptyset$; fix PRNG seed 42
2: **for** $r \in \{\mathrm{syn}, \mathrm{ant}, \mathrm{der}\}$ **do**
3:     **while** $|\mathcal{D}_r| < 1{,}000$ **do**
4:         sample distinct $A, C, E \in \mathcal{W}$
5:         $B \leftarrow \mathrm{uniform}(R_r(A)),\ \ D \leftarrow \mathrm{uniform}(R_r(C)),\ \ F \leftarrow \mathrm{uniform}(R_r(E))$
6:         **if** all six tokens are distinct and share no substrings **then**
7:             add $\big((A, B), (C, D), (E, F), r\big)$ to $\mathcal{D}_r$
8:         **end if**
9:     **end while**
10: **end for**
11: **return** $\bigcup_r \mathcal{D}_r$

---

The script rejects instances with surface overlap to prevent trivial pattern matching.

**Corpus statistics (length, part-of-speech, candidate-set sizes) are reported in Appendix A.**

### 2.3 Prompt construction

Each item is rendered as two support lines plus the query,

$$\mathtt{A\ :\ B\ \ C\ :\ D\ \ E\ :\ ?,}$$

followed by a strict "answer-only" guard (see `utils.py`). No chain-of-thought or natural-language instruction is provided.

### 2.4 Inference protocol

Models are queried with greedy decoding (temperature 0, `max_new_tokens`=2), forcing a single-token answer. This isolates relational induction from prompt-engineering effects.

### 2.5 Scoring

Let $\hat{F}$ be the normalised first token produced by the model. The scoring function is

$$s(\hat{F}, E, r) = \mathbf{1}\big[\hat{F} \in R_r(E)\big].$$

Overall accuracy is $\hat{p} = \frac{1}{3{,}000} \sum_{i=1}^{3{,}000} s_i$. Under a binomial model, $\mathrm{Var}[\hat{p}] = \frac{\hat{p}(1-\hat{p})}{3{,}000}$, so the 95% Wald interval equals $\hat{p} \pm 1.96\sqrt{\mathrm{Var}[\hat{p}]} \approx \hat{p} \pm 1.8\,\mathrm{pp}$.

Table 1: Accuracy on LAMBDA; 95% confidence half-width is $\pm 1.8$ pp.

| Model | Params | Overall | Syn | Ant | Der |
|---|---|---|---|---|---|
| Human Evaluation | — | 51.3% | 64.5% | 42.1% | 47.2% |
| | | | | | |
| GPT-2 medium (Radford et al., 2019) | 0.3B | 0.3% | 0.3% | 0.4% | 0.2% |
| Mistral-7B-Instruct (Jiang et al., 2023) | 7B | 4.9% | 4.5% | 7.0% | 3.2% |
| Llama-4 Scout-17B (Meta AI, 2025) | 17B | 40.4% | 42.1% | 46.9% | 32.1% |
| Llama-4 Maverick-17B (Meta AI, 2025) | 17B | 46.4% | 44.8% | 48.9% | 45.6% |
| | | | | | |
| GPT-4o[2] (OpenAI, 2024) | — | 31.3% | 23.7% | 37.8% | 32.4% |
| GPT-4.1[2] (OpenAI, 2025) | — | 32.2% | 23.4% | 38.0% | 35.3% |
| GPT-4.1 nano[2] (OpenAI, 2025) | — | 22.7% | 17.3% | 30.5% | 20.3% |
| Gemini 2.5 Pro[2] (Google DeepMind, 2025) | — | 24.2% | 22.5% | 25.6% | 24.6% |

## 2.6 System diagram

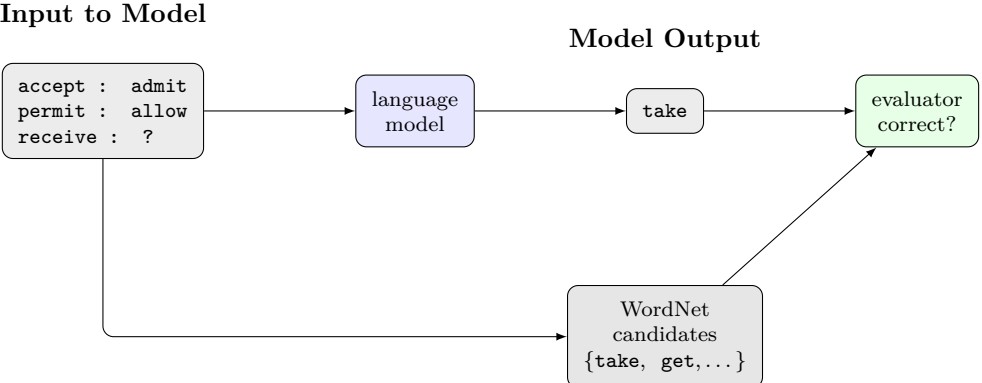

Figure 1: Scoring flow for an example synonym entry. Two support pairs define the hidden relation; the model outputs one word, and the evaluator checks membership in the WordNet-derived candidate set.

## 2.7 Output Collection

We report results for four open-source and four commercial models (0.1B–17B parameters), queried via the HuggingFace Inference API (Wolf et al., 2020) and the OpenAI API. Each run streams tokens and logs per-item correctness for aggregation.

Human evaluation was completed on a randomly selected subset of the dataset. 100 items were chosen from each category, and the evaluator (a native English speaker) was provided with the ability to determine word definitions.

# 3 Experiments and Analysis

## 3.1 Baselines

Table 1 shows the four open-source checkpoints and four commercial checkpoints we ran tests upon. Confidence bounds use the formula in Section 2.5.

---

[2] Parameter counts for these models are not publicly available.

**Comparison to human performance.** To contextualize the model results, we evaluated a human partic­ipant on a randomly selected subset of 300 analogies (100 per relation type). The human achieved an overall accuracy of **51.3%**, with scores of 42.1% on antonyms, 64.5% on synonyms, and 47.2% on derivations. No evaluated model—including the largest open or commercial systems—surpassed 0.5 accuracy in any category. This gap highlights the remaining challenges in analogical reasoning for today's language models, even under minimal-shot conditions.

### 3.2 Relation difficulty

Small models find antonyms easiest, likely because polarity cues (e.g. *hot–cold*, *increase–decrease*) are memo­rised during pre-training. Synonyms pose a similar challenge to derivations for the 17 B Maverick checkpoint, but derivations remain hardest for the 7 B Mistral model, suggesting greater morphological abstraction is needed. The noun-heavy skew in synonym and derivation queries versus the adjective tilt in antonyms (Fig­ure A.2 in Appendix A) aligns with this pattern: polarity adjectives are short, frequent, and more easily matched, whereas derivational morphology often requires longer stems or suffix manipulations.

### 3.3 Error taxonomy

A brief manual analysis of random failures reveals three dominant patterns. Let $R_r(q) = \{w_1, w_2, \ldots, w_n\}$ denote the candidate set of target words, from which the model must select one correct word for a given query $q$.

- **Identity Echo**: The model repeats the query token $q$ instead of generating a word from $R_r(q)$. For example, given $q = $ cat, the model outputs cat.

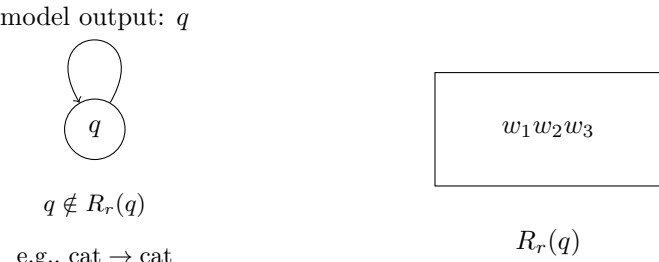

- **Surface Misfire**: The model applies an irrelevant form change to $q$ (e.g., pluralization), resulting in $q'$, which is not in $R_r(q)$. For example, given $q = $ cat, the model outputs cats.

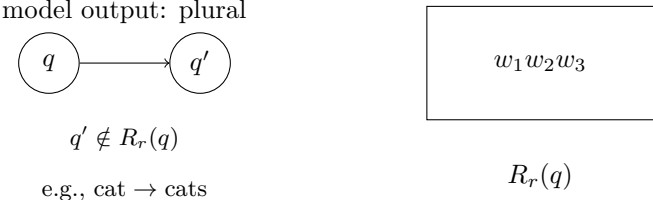

- **Semantic Drift**: The model generates $w$, which is semantically related to $q$ but not in $R_r(q)$, indicating a near-miss. For example, given $q = $ big, the model outputs large.

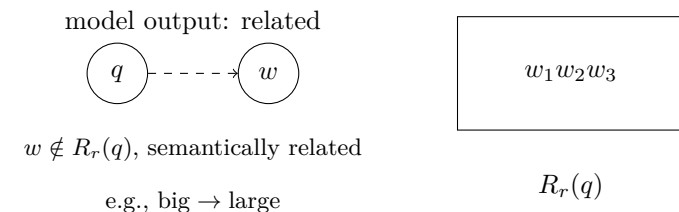

### 3.4 Length ablation

For SCOUT-17B (henceforth also referred to as Scout), accuracy *rises* with query-word length (Spearman $\rho = 0.52$; see Figure A.1 for the underlying length distribution). Short, high-frequency lemmas benefit from denser distributional evidence, but longer stems appear to support stronger relational abstraction once the model recognises the lexical pattern. Accuracy is stable to decoding temperature: results for $T = 0.1$ and $T = 0.2$ varied by at most $\pm 1\%$. A length-accuracy scatter plot is provided in the Discussion section (Figure 2), where this is explored further. The code used to compute the correlation can be seen in Appendix C.

## 4 Discussion

### 4.1 Overview of Empirical Findings

Our results trace a variable performance ladder on LAMBDA, spanning three orders of magnitude in parameter count. GPT-2 medium (0.3 B parameters) answers only nine of the 3,000 analogies, and even a modern 7 B instruction-tuned checkpoint solves fewer than 5% of items. In contrast, the two Llama-4 17 B variants both surpass the 40% threshold, indicating that large-scale pre-training and instruction tuning are prerequisites for lexical-level relational abstraction. The $\pm 1.8\,\mathrm{pp}$ confidence bands reported in Table 1 guarantee that these gaps are statistically robust.

While the raw numbers confirm long-standing observations that analogy is brittle for small models (Drozd et al., 2016), they also coincide with evidence that seemingly "emergent" behaviours surface abruptly once models cross a certain scale (Opiełka et al., 2025). Recent work cautions, however, that such break-points may partly reflect metric granularity rather than genuine phase transitions (Schaeffer et al., 2023). Taken together, our baselines position LAMBDA as a sensitive analysis of the middle-to-upper portion of today's model zoo, capable of separating near-state-of-the-art open-source checkpoints that look indistinguishable on headline leaderboards.

### 4.2 Relation-wise Performance

Table 2: POS (part of speech) counts for query words (Penn tags grouped).

| POS | Synonym | Antonym | Derivation | Total |
|---|---|---|---|---|
| Noun | 807 | 542 | 764 | 2 113 |
| Verb | 6 | 11 | 9 | 26 |
| Adjective | 88 | 216 | 109 | 413 |
| Adverb | 26 | 83 | 10 | 119 |
| **Total** | 1 000 | 1 000 | 1 000 | 3 000 |

As Table 2 shows, synonym and derivation queries are overwhelmingly noun-heavy, whereas antonym queries skew heavily toward adjectives.

**Antonyms dominate early.** Across all checkpoints, antonym questions are the easiest slice (216 adjective queries). Even GPT-2, a comparatively primitive model, solves a handful here, and Mistral-7B reaches the 7% mark. This mirrors psycholinguistic evidence that polarity pairs (e.g. *hot–cold*) are acquired early and occur disproportionately in text, giving models strong distributional cues (Drozd et al., 2016). Counter-fitting work further shows that antonym relations are linearly separable in embedding space (Mrkšić et al., 2016), which may explain their accessibility to small LMs.

**Synonyms tread water.** Performance on synonym analogies grows more slowly with scale: Scout improves over Mistral by 38 pp, yet Maverick gains only another 2 pp. Reflecting their noun-dominated com-

position (807 nouns), synonyms suffer from lexical ambiguity—WordNet synonym sets often span multiple senses, forcing disambiguation from minimal context.

**Derivations are the long tail.** Derivational morphology remains hardest overall. Although derivations also exhibit a strong noun majority (764 nouns), they include more verbs (9) and adjectives (109), introducing added morphological variation. The gap between synonyms and derivations flips sign between Scout and Maverick, hinting that beyond roughly 10 B parameters, models begin to learn affix-level regularities (Vylomova et al., 2017). Yet Maverick still misses more than half of the derivational items, echoing recent morphology-specific evaluations that document sizeable headroom even for GPT-class systems (Romanov & Khusainova, 2023).

**A unifying lens.** The sub-slice scores suggest a competence hierarchy:

$$\text{Antonymy} > \text{Synonymy} \approx \text{Derivation (small models)}, \quad \text{Antonymy} \approx \text{Derivation} > \text{Synonymy (large models)}.$$

Early gains reflect frequency and surface cues; later gains reflect emerging morphological abstraction. Further ablating part-of-speech (Figure A.2) and candidate-set size (Table A.3) should disentangle frequency from compositional complexity.

### 4.3 Length Ablation Continued

We next examine how query-word length modulates relational abstraction in Scout-17B, grounding our analysis in information-theoretic and cognitive principles. Zipf's law of abbreviation predicts an inverse relationship between word length and frequency, implying that longer lemmas should carry disproportionately high information and thus be easier to retrieve or analogize in context (Zipf, 1935; i Cancho & Solé, 2003; Zipf, 1949). Surprisal theory further posits that the processing difficulty of a word scales with its negative log-probability; thus, longer and less frequent words are associated with higher surprisal and supply stronger contextual cues, which may facilitate analogical mapping under minimal support (Piantadosi et al., 2011; Shannon, 1951). Morphological complexity is a critical variable in neural language models: affix-rich corpora promote the emergence of internal representations that exploit stem–suffix regularities, leading to accuracy boosts for longer stems once a sufficient parameter scale is reached (Mu & Tomos, 2023; Linssen & Rogers, 2022; Piotte et al., 2021; Yang et al., 2024).

Empirically, we observe a moderate positive Spearman correlation ($\rho = 0.52$, Figure 2) when examining the full range of lengths, confirming that Scout-17B more reliably solves longer-word analogies. However, many theoretical and empirical works warn against conflating effects due to noise from sparsely sampled lengths or heavy-tailed distributions in language data (Norris & Cutler, 2021; Linssen & Rogers, 2022; Piantadosi et al., 2011). Recomputing the correlation after removing extreme-length outliers (i.e., restricting to lengths with at least five examples) yields a markedly stronger Spearman coefficient of 0.68, highlighting that the positive association between length and accuracy is not merely a product of noise or spurious structure at distributional extremes.

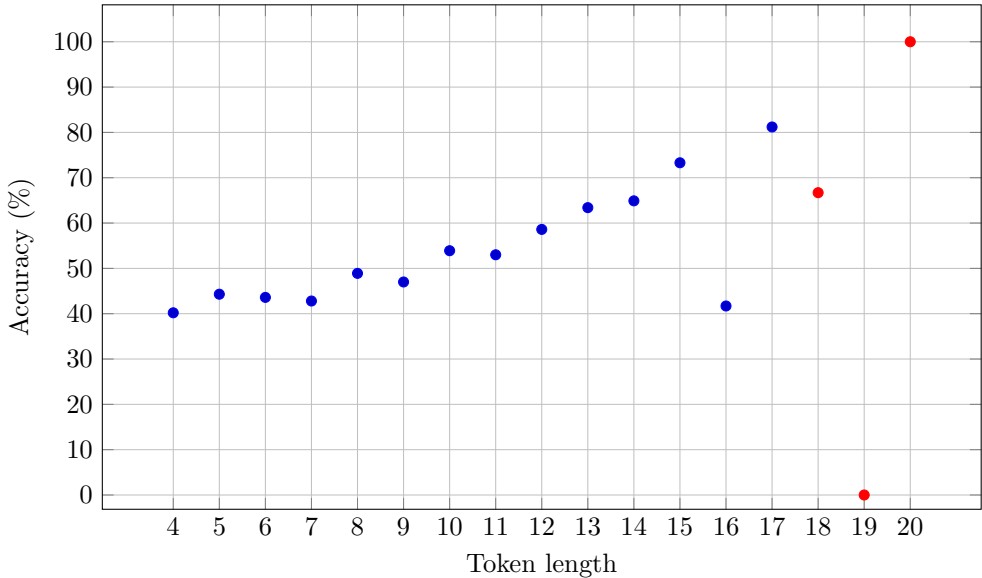

Figure 2: SCOUT-17B accuracy by query length ($n < 5$). Spearman $r = 0.52$. Outliers are marked in red.

A complementary theoretical curve (Figure 3) illustrates the idealized Zipfian decline in word frequency with rank, contextualizing the deviation of empirical analogy accuracies from a pure power law (Zipf, 1949; 1935; i Cancho & Solé, 2003). These patterns reinforce classic findings that information per character in natural language scales sublinearly but cumulatively with length, yielding richer distributional cues for models to exploit (Piantadosi et al., 2011; Koehn & Knowles, 2017).

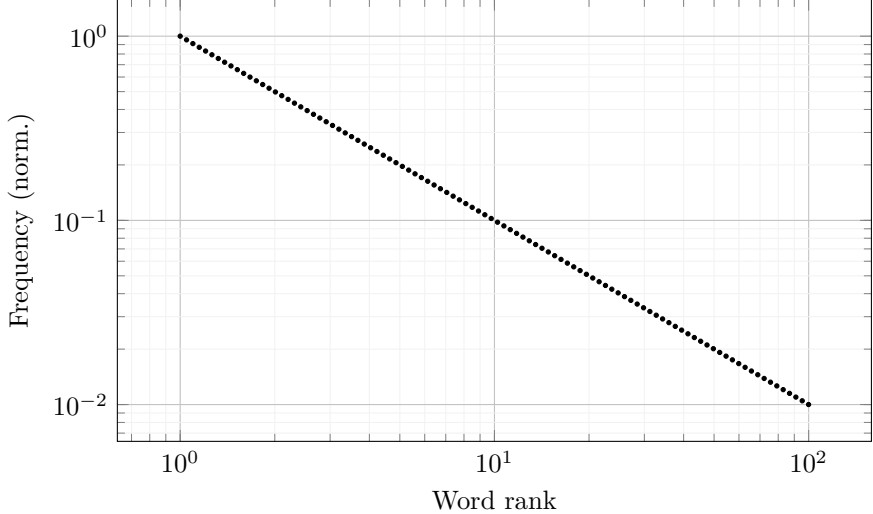

Figure 3: Idealized Zipf's law ($f \propto r^{-1}$) on log–log axes with logarithmically spaced samples.

Recent ablation studies and theoretical models converge on the view that for SCOUT-17B, longer lemmas not only transmit greater information content but also activate richer morphological cues, yielding substantially higher analogy accuracy. Character-aware and segment-guided models achieve improved robustness to tokenization variance and generalize better for long or morphologically complex words (Mu & Tomos, 2023; Yang et al., 2024; Clark et al., 2022). Moreover, neural architectures exhibit "token bottleneck" effects, wherein short or ambiguous words disproportionately reveal model compositional weaknesses, while

affix-aware pretraining helps mitigate these effects by encouraging the use of structure across variable-length spans (Turner & Williams, 2023; Piotte et al., 2021).

Nevertheless, the gains associated with length are not strictly monotonic: for example, accuracy at lengths 16 and 17 deviates from the prevailing trend, likely due to idiosyncratic word-type effects or low sample size. We recommend that future correlation analyses be reported both with and without extreme outliers to ensure replicable conclusions regarding the scaling properties of analogy accuracy.

## 4.4 Comparison to Prior Analogy Benchmarks

A growing body of research has relied on lexical analogy benchmarks to evaluate semantic and morphological abstraction in word and language models. The most prominent datasets in English include the Google Analogy Test Set (GAT; Mikolov et al. 2013b), BATS (Gladkova et al. a), and WordRep (Gao et al. 2014). Each targets a broad set of lexical or morphosyntactic relations, with varying coverage, distractor construction, and part-of-speech composition.

Compared to our LAMBDA collection, which emphasizes minimal-shot analogical inference across synonymy, antonymy, and derivation, these earlier benchmarks typically present higher-context, fixed-format prompts (e.g., `man:king::woman:??`), and often conflate surface analogy with broader lexical similarity. Recent work has critiqued their redundancy, predictability, and ceiling effects for large models, particularly for analogy categories dominated by frequent words or direct word-form changes (Bouraoui et al., 2020).

Our findings reinforce these critiques: even high-performing instruction-tuned LLMs that approach ceiling on legacy analogy sets reveal substantial headroom when faced with our minimal-shot, mixed-relation challenge. The introduction of derivational categories in particular exposes compositional gaps not surfaced by prior datasets (Gladkova et al., a; Drozd et al., 2016; Vylomova et al., 2017). We therefore position LAMBDA as a more granular probe of genuine relational abstraction in contemporary LMs

## 4.5 Scaling Law Extrapolation

Classic scaling laws predict log-linear improvements in accuracy with increasing parameter count (Kaplan et al., 2020). However, our results depart sharply from this trend. As Table 1 shows, Llama-4 Maverick-17B achieves the highest score (46.4%), substantially outperforming both GPT-4o and GPT-4.1 (31.3% and 32.2%, respectively), even though the latter are widely presumed to be much larger and generally stronger models.

This discontinuity suggests that model size alone is a poor predictor of relational analogy induction; data curation, architecture, or specific training regimes may matter more than scale in this setting. Our findings echo recent work questioning the universality of scaling laws and the nature of "emergent" abilities in LLMs (Schaeffer et al., 2023). Overall, scaling up does not guarantee robust lexical relational abstraction, and highly-tuned open-source models may surpass commercial systems on tasks outside headline benchmarks.

## 4.6 Broader Impact

Systematic analogical reasoning is a central component of human language understanding, with broad implications for tasks such as scientific discovery, education, and knowledge transfer. By developing LAMBDA, a minimal-shot lexical analogy benchmark, we provide researchers and practitioners with a more targeted way to assess whether language models move beyond surface pattern matching toward genuine relational inference. While the current results reveal that even state-of-the-art models have not mastered these capabilities, a clearer diagnostic benchmark can help focus community efforts on genuine abstraction rather than superficial accuracy.

This work may also inform future research in linguistic theory, cognitive modeling, and educational technology by clarifying which relational skills remain out of reach for current systems. At the same time, exposing specific weaknesses in analogy-making could help mitigate the risk of overestimating language model generalization in downstream applications such as question answering, knowledge graph completion, or scientific

information extraction. More rigorous evaluation of relational abstraction is a step toward safer and more interpretable AI, especially as models are deployed in high-stakes or high-impact language tasks.

## Limitations and Future Directions

The present version of LAMBDA provides a targeted yet necessarily partial lens on analogical reasoning in language models. The design choices—English-only vocabulary, WordNet-derived relations, strict single-token scoring, and uniform two-shot prompts—were made to ensure interpretability, reproducibility, and statistical rigor. At the same time, these constraints define the edges of what the benchmark currently measures and suggest a broad pathway for future versions of our dataset and use in general research.

One central limitation is linguistic scope. LAMBDA is English-only and restricts its queries to lemmas covered by WordNet, a resource that, while comprehensive, does not encompass the diversity of natural language in either morphology or lexical innovation. This leaves open important questions about model performance in morphologically rich languages, low-resource settings, or informal registers. For example, would models exhibit the same patterns of relational abstraction if tested on Finnish, Turkish, or code-switched datasets? Developing multilingual or cross-lingual extensions of LAMBDA would allow for systematic investigation of the transferability and universality of analogical reasoning.

The composition of the candidate sets reflects a further bias: WordNet favors formal, established vocabulary and tends to under-represent colloquial, dialectal, or neologistic forms. As a result, models may struggle more with real-world analogy problems that involve emerging or domain-specific terms, and observed difficulty levels may partly reflect this sampling. Addressing lexical coverage bias could involve augmenting the benchmark with dynamic corpora or alternative lexical resources, ensuring the analogies span both standard and non-standard language.

Another notable constraint lies in the evaluation protocol. The current benchmark accepts only single-token, exact-match responses, which, while enabling clear scoring, underestimates true competence in cases where valid paraphrases or multiword expressions are possible but not explicitly accepted by the gold standard. Multi-token predictions, semantic equivalence scoring, and more flexible matching criteria could provide a fuller picture of the analogical capabilities of both models and humans. Such extensions would also allow for evaluation of compositionality beyond the lexical level, capturing when models generate plausible but novel analogical mappings. Future versions of this dataset could also rely on hypernyms[3] and hyponyms[4] in evaluation.

The uniform prompt format (three isolated lines with two support pairs) was chosen to minimize the confounding effects of prompt engineering and context, isolating the minimal setting for relation induction. Yet, this also omits potentially beneficial effects of richer prompts, additional examples, or explicit reasoning instructions. Future experiments could explore the full space of prompt formats, including varying few-shot and many-shot, to map how different cues and contextual signals affect analogical abstraction.

In focusing on individual lemmas, the current benchmark necessarily omits phrase-level, sentential, or multimodal analogies. Human analogical reasoning often operates over much richer and more structured inputs—comparing phrases, diagrams, or event schemas, for instance. Extending the benchmark to contextual or phrase-level analogies, or to analogies that bridge language and vision, would provide a more complete and ecologically valid assessment of systematic generalization.

It is also worth noting that, while we correlate accuracy with model scale and training regimen, the present analysis does not causally disentangle which aspects of the pretraining data, objectives, or architectural features drive analogical competence. Controlled ablation studies—varying one factor at a time—could reveal, for instance, whether explicit morphological annotation, targeted relation pretraining, or architectural innovations such as affix-aware embeddings lead to greater abstraction or generalization.

Taken together, these limitations define a clear roadmap for future work. Multilingual extensions, relaxed and graded evaluation protocols, prompt and context ablations, richer analogy types, and controlled causal

---

[3] A hypernym is a broader term that includes more specific words; e.g., *animal* is a hypernym of *dog*.
[4] A hyponym is a specific word under a broader term; e.g., *dog* is a hyponym of *animal*.

studies all offer promising directions. In particular, building benchmarks that span languages, admit flexible outputs, and invite both fine-grained and broad generalization will enable a deeper and more nuanced understanding of how, when, and why language models develop analogical reasoning. Addressing these frontiers will not only advance evaluation methodology but also shed light on the fundamental mechanisms that support abstraction in artificial and natural learners alike.

## 5 Conclusion

This study introduces LAMBDA, a benchmark designed to evaluate few-shot lexical analogical reasoning in language models. We constructed a dataset of 3,000 relation-hidden analogies spanning synonyms, antonyms, and derivational morphology, and used it to systematically assess both open and closed models across a range of scales. Our experiments reveal a steep performance gradient: no model surpassed 50% accuracy overall or within any relation category. Our length-based analysis and manual error review confirm that major challenges persist, particularly for synonyms and derivations, and that systematic generalization remains unresolved, even in today's strongest models.

Although language models have advanced quickly, there is still a substantial gap between surface pattern recognition and true reliable analogical reasoning. LAMBDA enables more detailed comparison between models that otherwise appear similar on headline benchmarks and motivates the development of more specialized evaluation protocols. The limitations discussed here point to several concrete directions for future research, including creating multilingual datasets, allowing flexible multi-token answers, exploring more diverse prompt formats, and conducting targeted ablations to better understand the mechanisms behind relational inference.

Overall, our results emphasize the need to go beyond aggregate performance metrics and toward more rigorous evaluation of abstract relational reasoning. By releasing LAMBDA, we hope to enable future work that addresses these limitations and brings language models closer to robust and systematic language understanding. Our findings highlight a persistent gap between human-level and model-level analogical reasoning, underscoring the need for targeted benchmarks like LAMBDA to drive progress toward more systematic generalization in language models.

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

# Appendix A  Dataset Statistics

## A.1 Token-length distribution

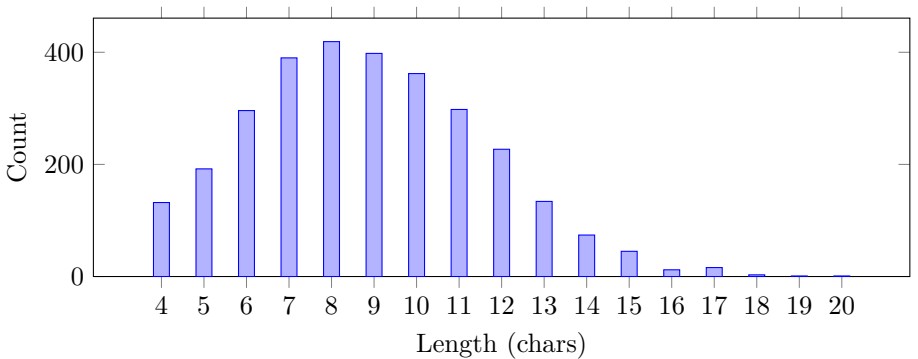

Figure A.1: Histogram of query-word lengths. Median = 9, 90th percentile = 12.

Table A.1: Counts of query-word lengths.

| Length | 4 | 5 | 6 | 7 | 8 | 9 | 10 | 11 | 12 | 13 | 14 | 15 | 16 | 17 | 18 | 19 | 20 |
|---|---|---|---|---|---|---|---|---|---|---|---|---|---|---|---|---|---|
| Count | 132 | 192 | 296 | 390 | 419 | 398 | 362 | 298 | 227 | 134 | 74 | 45 | 12 | 16 | 3 | 1 | 1 |

Table A.2: Descriptive statistics for query-word length.

| Count | Mean | Median | 90-pct | Min | Max |
|---|---|---|---|---|---|
| 3 000 | 8.85 | 9 | 12 | 4 | 20 |

Ten percent of queries contain 13–20 characters, adding morphological variety that can hinder surface memorisation.

## A.2 Part-of-speech breakdown

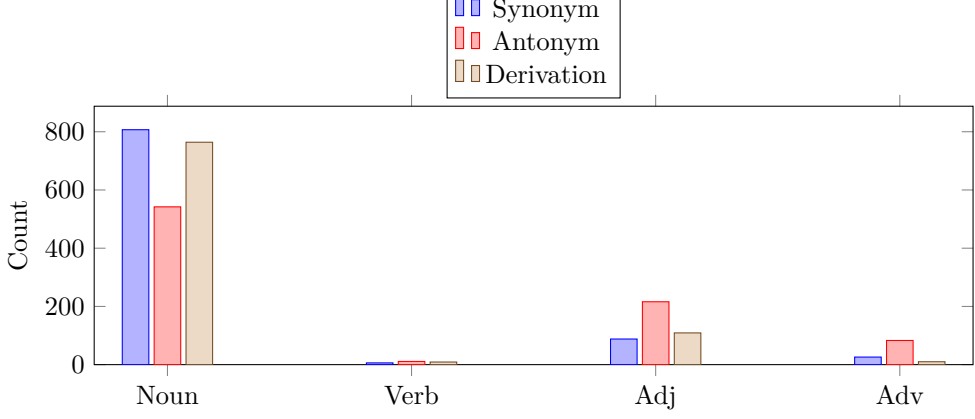

Figure A.2: POS distribution per relation.

Synonym and derivation queries are noun-heavy, while antonyms skew toward adjectives, mirroring WordNet polarity pairs such as *hot–cold*.

### A.3 Candidate-set sizes

Table A.3: WordNet candidate-set size $|R_r(E)|$ for query lemmas.

| Relation | Mean | St Dev | Min | Max |
|---|---|---|---|---|
| Synonym | 3.93 | 4.96 | 1 | 54 |
| Antonym | 1.35 | 0.71 | 1 | 5 |
| Derivation | 3.53 | 4.13 | 1 | 62 |

Antonym queries are nearly single-choice; synonym and derivation queries present much larger target sets, demanding stronger relational inference.

## Appendix B    Generation and Analysis Scripts [5]

Listing 1: Deterministic item-generation script

```python
import json, random
from nltk.corpus import wordnet as wn

def get_synonyms(w):
    s = set()
    for syn in wn.synsets(w):
        for l in syn.lemmas():
            if l.name().lower() != w.lower():
                s.add(l.name().replace('_',' '))
    return list(s)

def get_antonyms(w):
    s = set()
    for syn in wn.synsets(w):
        for l in syn.lemmas():
            for a in l.antonyms():
                if a.name().lower() != w.lower():
                    s.add(a.name().replace('_',' '))
    return list(s)

def get_derivations(w):
    s = set()
    for syn in wn.synsets(w):
        for l in syn.lemmas():
            for d in l.derivationally_related_forms():
                if d.name().lower() != w.lower():
                    s.add(d.name().replace('_',' '))
    return list(s)

def pick_related(word, func):
    lst = func(word)
    return None if not lst else random.choice(lst)

words = [w for w in set(wn.all_lemma_names())
         if w.isalpha() and w.islower() and len(w) > 3]

random.seed(42)
dataset = []

for func, label in [(get_synonyms,"synonym"),
```

---

[5]Full code will be released on GitHub upon publication.

```
41                    (get_antonyms,"antonym"),
42                    (get_derivations,"derivation")]:
43      cnt = 0
44      while cnt < 1000:
45          chosen = set()
46          triples = []
47          for _ in range(3):
48              tries = 0
49              while tries < 50:
50                  c = random.choice(words)
51                  if c not in chosen:
52                      r = pick_related(c,func)
53                      if r and r not in chosen:
54                          triples.append((c,r))
55                          chosen.update((c,r))
56                          break
57                  tries += 1
58          if len(triples) != 3: continue
59          (A,B),(C,D),(E,F) = triples
60          if len({A,B,C,D,E,F}) < 6: continue
61          dataset.append({"few_shot":[{"input":A,"output":B},
62                                      {"input":C,"output":D}],
63                          "question":f"{E} : ?",
64                          "relation":label})
65          cnt += 1
66
67  with open("dataset/lexical_dataset.jsonl","w",encoding="utf-8") as f:
68      for entry in dataset:
69          f.write(json.dumps(entry)+"\n")
```

The following script computes per-length accuracies and correlation:

Listing 2: Length-wise accuracy and Spearman correlation

```
1   import pandas as pd
2   from scipy.stats import spearmanr
3
4   data = [
5       (4, 40.2), (5, 44.3), (6, 43.6), (7, 42.8), (8, 48.9),
6       (9, 47.0), (10, 53.9), (11, 53.0), (12, 58.6), (13, 63.4),
7       (14, 64.9), (15, 73.3), (16, 41.7), (17, 81.2), (18, 66.7),
8       (19, 0.0), (20, 100.0)
9   ]
10
11  df = pd.DataFrame(data, columns=["length", "accuracy"])
12  r, temp = spearmanr(df["length"], df["accuracy"])
13  print(f"Spearman r = {r:.2f}")
```

