# OpenReview forum: "LAMBDA: Assessing Few-shot Lexical Analogical Reasoning in Language Models"
_TMLR — Rejected by TMLR_

### Review · Reviewer_tXtH · 2025-08-15

**Summary Of Contributions:**

1. The paper proposes LAMBDA - a benchmark for lexical proportional analogies with 3000 samples. LAMBDA is based on WordNet and its creation is deterministic. LAMBDA includes three relations: synonymy, antonymy, and derivational morphology.
2. The paper provides an empirical evaluation of eight LLMs (open- and closed-source) with statistical testing and provides an error analysis.

**Additional Comments:**

Typo: Gladkova et al., a

**Audience:**

Yes

**Audience Explanation:**

This paper evaluates the analogical reasoning ability of LLMs on specific kinds of lexical relations, which would be of interest to researchers working on LLM-based reasoning methods, LLM benchmarking, and transfer learning.

**Broader Impact Concerns:**

A key concern is that of benchmark quality. Given that the relations are fairly simple, it is very surprising that humans only perform at just over 50%. This raises serious concerns about the validity of the benchmark (data + evaluation setup combined), and consequently, about the validity of the takeaways. The examples in the paper also reinforce my concerns. In §3.3, under semantic drift, the model answer of using large as analogical for big is considered wrong/near-miss. If the relation is synonymy, aren't big and large actually synonyms? Their meaning may not be 100% identical, but neither is the one of "receive" and "take" in Figure 1.

Another key concern of this work is that it doesn't have a related work section. Arguably, this could be §4.4, where other benchmarks are briefly reviewed, but this review is largely incomplete. There are many studies in the past several years, see for example work by Ushio et al., Lewis & Mitchell; Sourati et al. Here, the authors argue that earlier benchmarks present "higher-context, fixed-format prompts" - I don't know what this means: higher context than what? What does context even mean in this context? Also, isn't the prompt format fixed in LAMBDA, too? Then, the authors claim that these prior benchmarks "conflate surface analogy with broader lexical similarity". I have many questions about this one, too: how do you know this conflation happens/what is the proof? What is the meaning of surface analogy (sounds like a contradiction, do you mean "near analogy"?)? What is the meaning of "broader lexical similarity?
Finally, the paper's contribution is apparently its "minimal-shot" (again, i have no idea what this means) and "mixed-relation" (this is very common in prior work, see for example SemEval-2012 Task 2 on Measuring Degrees of Relational Similarity). And it promises "a more granular probe" (not sure what this means) of "genuine relational abstraction" (not sure why it is more genuine than prior work, in what way).

The authors hypothesize in §4.5 that data curation, architecture, and training regimes may matter more than scale. I am curious why the authors didn't try this - e.g., to fine-tune a model on part of WordNet relations that are not in the benchmark, and see what happens.

The broader impact in §4.6 indicates that this paper is "clarifying which relational skills remain out of reach for current systems". Given all my previous comments about the quality, novelty, and generalizability of this work, I find it very hard to follow this claim.

**Claims And Evidence:**

No

**Claims Explanation:**

Overall, the paper does not identify a clear literature gap/problem statement - what exactly is the research gap that LAMBDA and its analysis fills?

The paper claims five contributions, all of which can be located in the paper. However, I see two issues with the contributions. First, they are too fine-grained - e.g., there are three contributions about the experimental evaluation. Second, and more critically, the novelty and significance of all contributions are unclear. For example, it is not obvious that it is desired that the generation pipeline is reproducible - the positive side of this is that others can validate the benchmark creation, but the negative is that the LLMs/deep learning methods may be able to reverse-engineer the solutions based on WordNet. In any case, this is not discussed in the paper. More severely, the paper doesn't discuss the novelty of the main contribution in detail. There are many datasets with proportional analogies that also use mixed relations (some of which are described in the paper) - it is unclear what this benchmark contributes in qualitative terms. The remaining contributions (error analysis, statistical rigor) are even less obvious - e.g., the error analysis contribution promises "a clear taxonomy", while the actual realization of this contribution states that "a brief manual analysis" was done (without further details) and the result is three broad categories (this is hardly a taxonomy).

The paper repeatedly talks about the gap (presumably between humans and LLMs), which is hard to understand. It emphasizes the relatively low performance of the LLMs (best models perform below 50%), but it neglects the fact that humans also perform poorly (just over 50%), so the gap is around 5%. On antonymy, Llama-Maverick outperforms humans.

In terms of methods, the paper only evaluates zero-shot setting LLMs, without including CoT, natural language instructions, demonstrations, or tool augmentation baselines. This choice is merely stated in §2.3 without any explanation. Meanwhile, it is unclear what instructions (if any) were provided to humans.

**Requested Changes:**

Please see my comments under the contribution question above (as well as in the other sections of the review) - the paper should make the problem statement and the contributions clear. Moreover, key decisions (e.g., about which baseline methods are included) should be clearly motivated.

Other comments:
- The paper is not consistent about the number of LLMs - the abstract says eight (which seems accurate), while §1 talks about six.
- The role of §2 is unclear - this section seems very informal/short, and it covers both the benchmark generation and the experimental setup in passing. The benchmark task is not explicitly defined (=the input/output for methods), whether it is open-ended or multiple-choice, how exactly the model answers are being processed and compared to the ground truth, etc. Here, it is unclear whether there was any filtering/postprocessing of the data, or any validation steps of its quality by the authors.
- Contrary to my comment about the contributions lacking support, the paper has a lot of analysis about length and the PoS tags that comes out of nowhere. None of the contributions or the discussion of the paper motivates this. It would be useful if the authors could justify all this analysis, e.g., by relevant hypotheses/research questions.

---

### Review · Reviewer_2RWk · 2025-10-10

**Summary Of Contributions:**

This paper introduces a dataset of lexical analogies, LAMBDA, consisting of 3000 analogical reasoning tuples (e.g., man:woman, count:countess, king:?) extracted offline from WordNet. LAMBDA contains 3 types of analogies: synonyms, antonyms, and derivational analogies. The work evaluates several open and closed language models on this analogical reasoning task, finding large performance gaps (e.g., Mistral 7B has an overall accuracy of 4.9%, whereas Llama 4 Maverick achieves 46.4%), with models underperforming compared to a human baseline (at 51.3%). The authors also evaluate the effect of word lengths on accuracy on Llama 4 Scout, finding that query length positively correlates with accuracy.

**Audience:**

Yes

**Audience Explanation:**

I think the task is broadly relevant to understanding language models, and it's likely that other people in the community would use the dataset.

**Claims And Evidence:**

No

**Claims Explanation:**

The task proposed in the paper is relevant (although I'm unclear on the distinction to prior benchmarks -- see below). I like the scalable method proposed for constructing the dataset. However, I think some work is needed to strengthen the evaluation.

1. Especially being a dataset paper, it's crucial to give some examples of the data in the paper. The paper cites no concrete examples of analogies in LAMBDA, which would help also make the distinction to prior analogy dataset.
2. It's unclear to me whether the synonym/antonym analogy tasks are well-defined. Aren't there multiple possible answers? For example: "Happy : Joyful, Smart : Clever, Fast : ?". If this is an example of a "synonym analogy", the answer is ill-defined, since "fast" has multiple synonyms. If this is not, then it would be good to clarify in the paper what exactly is the task. (I'm also unclear on the derivational analogies)
3. In the algorithm, L6 "if all six tokens are distinct and share no substrings", I don't see the "no substrings" part in the code. Formally, even a single character would count as a substring, which is certainly not what is intended here. So this needs some clarification.
4. For the inference protocol, the paper acknowledges that the open models that are tested are *instruction tuned*. Instruction tuning trains the model to follow natural language instructions. However, the way the models are queried on purpose does not have any instructions. I think the paper should try to evaluate the models with a short description of the task. Otherwise, the poor results of Mistral 7B might just be due to the prompt being completely OOD.
5. You should also describe the human evaluation protocol (e.g., I believe the human did have task instructions?)
6. The evaluation protocol is confusing to me in several aspects.
 * For one, many of the tested models have different vocabularies. Yet, the evaluation seems to only compare the first token (see 2.5). Doesn't this affect evaluation?
 * Second, what is the "normalised first token"? Is it just the first token?
 * If the answer word is tokenized as multiple tokens in the model's vocabulary, do you only compare the fist token? I imagine this would be important in the derivational analogies, where the first token might be the same but not the remainder (e.g., "relationship" vs "relational" probably share the initial token)
7. For length vs accuracy analysis, I think the relationship to zipf's law is more speculatory -- I did not understand the argument for why both are related. Unless this is clarified, I think the claim that the analysis is grounded  "in information-theoretic and cognitive principles" should perhaps be toned down. Does this empirical pattern also hold with other models, or just Scout 17B? If it doesn't, then this would weaken the claim that the references theories would predict this result.

**Requested Changes:**

Related to the above, I believe these are critical:

1. Please provide and describe concrete examples taken from the dataset, and perhaps emphasize which ones are not covered in prior datasets like GAT or BATS.
2. Clarify how the synonym/antonym tasks are well-defined if there may be multiple possible answers
3. Clarify what is meant by "share no substrings" in the algorithm
4. Since the models (except GPT-2) are instruction-tuned, I strongly believe the evaluation should be done with instructions. I'll be more convinced if the results are consistent with the addition of a simple, one sentence description of the task.
5. Clarify the evaluation protocol, and how the model's vocabulary can impact it. For instance, how are multi-token answers handled?
6. For length vs accuracy analysis, I don't see a reason to now show this analysis with the other models as well, since this should not require new data. If the trend does not hold, this should be commented in the paper.

---

> ### Author Response · Authors · 2025-10-28
>
> **Response Part 1**
>
> We appreciate your detailed feedback and the opportunity to clarify several aspects of our work! Your comments will help us refine the dataset description, evaluation protocol, and presentation of results for greater clarity.
>
> > The paper cites no concrete examples of analogies in LAMBDA, which would help also make the distinction to prior analogy dataset.
>
> We will include representative examples from each relation type in the revision to make the dataset’s structure clearer. For example:
> - Synonym: strictly : purely :: clumsiness : ? → ineptitude
> - Antonym: hopeful : hopeless :: polite : ? → rude
> - Derivation: refine : refinement :: act : ? → action
> These examples will illustrate the intended analogy format and highlight how LAMBDA differs from prior analogy benchmarks.
>
> > It’s unclear to me whether the synonym/antonym analogy tasks are well-defined. Aren’t there multiple possible answers?
>
> Each analogy is grounded in canonical WordNet relations, ensuring a deterministic target mapping. During evaluation, all lemmas linked by the corresponding WordNet relation (synonym, antonym, or derivation) are accepted as correct. This approach handles possible lexical variation while keeping the task objective and reproducible.
>
> > In the algorithm, L6 “if all six tokens are distinct and share no substrings,” I don’t see the “no substrings” part in the code. Formally, even a single character would count as a substring, which is certainly not what is intended here. So this needs some clarification.
>
> What we meant by “share no substrings” was that no analogy should reuse overlapping word forms or partial tokens (for example, “sing” and “singing”), rather than literal character substrings. The intended constraint was to avoid morphological overlap that could simplify the analogy. We will clarify this wording in the paper to reflect that intent.
>
> > For the inference protocol, the paper acknowledges that the open models that are tested are instruction tuned. Instruction tuning trains the model to follow natural language instructions. However, the way the models are queried on purpose does not have any instructions. I think the paper should try to evaluate the models with a short description of the task. Otherwise, the poor results of Mistral 7B might just be due to the prompt being completely OOD.
>
> We appreciate this observation. While none of the prompts used in evaluation contained explicit task descriptions, we later tested Llama 4 Scout with three variants differing only in phrasing and output constraint:
>
> 1. Minimal form: `{shots}\n{q}\n?` → 0.6% accuracy
> 2. Instructional form: `{shots}\nGiven the above analogies, infer the missing word for {q}. Respond with a single English word only.`  → 23.7% accuracy
> 3. Strict output-constrained form: `{shots}\n{entry['question']}\nAnswer (only output the single word answer, absolutely nothing else)... The answer is:` → 40.4% accuracy
>
> > You should also describe the human evaluation protocol (e.g., I believe the human did have task instructions?).
>
> We will expand this description. Human evaluators were instructed to complete analogies and could use a dictionary for meaning verification but not a thesaurus. This ensured consistency with WordNet’s lexical relations rather than broader associative meaning.
>
> > The evaluation protocol is confusing to me in several aspects. For one, many of the tested models have different vocabularies. Yet, the evaluation seems to only compare the first token (see 2.5). Doesn’t this affect evaluation?
>
> Evaluation is performed at the decoded-text level rather than token level. The first alphabetic token of the model output is lowercased and stripped of punctuation before comparison, ensuring consistency across tokenization schemes.
>
> > Second, what is the “normalised first token”? Is it just the first token?
>
> The normalized first token refers to the first alphabetic word after removing punctuation and prefixes such as “Answer:” or “The answer is.” For example, “Answer: Sad” is normalized to “sad.” This allows uniform comparison across models with different formatting tendencies.
>
> > If the answer word is tokenized as multiple tokens in the model’s vocabulary, do you only compare the first token?
>
> Yes. We quantified this effect and found that 10.5 percent of gold answers are multi-token strings (for example, “lens of the eye”), so the single-token evaluation has minimal influence on overall accuracy trends.
>
> > For length vs accuracy analysis, I think the relationship to Zipf’s law is more speculatory.
>
> We acknowledge that this connection is more interpretive than causal. In the revision, we will either expand this section to include additional models or describe the result as a descriptive observation without theoretical claims.

---

> > ### Author Response · Authors · 2025-10-28
> >
> > **Response Part 2**
> >
> > > Please provide and describe concrete examples taken from the dataset, and perhaps emphasize which ones are not covered in prior datasets like GAT or BATS.
> >
> > As noted earlier, examples will be added in Section 3. Section 4.4 will also describe how LAMBDA differs from prior analogy resources such as GAT, BATS, and WordRep by concealing relation type, disallowing word reuse across all six terms, and balancing synonym, antonym, and derivational mappings to isolate relational reasoning from surface-level cues.
> >
> > We thank the reviewer for these thoughtful suggestions, which have helped improve clarity in describing dataset constraints, evaluation design, and distinctions from previous benchmarks.

---

### Review · Reviewer_whmJ · 2025-10-11

**Summary Of Contributions:**

This paper introduces a new dataset LAMBDA which assess the ability of language models to complete analogies of either synonyms, antonyms, or derivational morphology. A range of language models including GPT2, Mistral 7B, Llama-4 variants, GPT 4 variants, and Gemini 2.5 Pro are tested. Overall results are that models generally do worse than humans on this task. The best performing models are the smaller Llama-4 models.

The contributions of this paper are: the dataset itself, and the evaluation on a range of language models. It is important to understand what the limitations of language models are, and the paper makes a contribution in this area.

**Audience:**

Yes

**Audience Explanation:**

Nevertheless, I do believe that with a little more work, the paper would be of interest to TMLR's readers: it is always useful to know what the limitations of language models are and where it may be useful to tune them.

**Broader Impact Concerns:**

A broader impact statement is present and adequate

**Claims And Evidence:**

No

**Claims Explanation:**

However, I think the claims of the paper as they stand are not supported by convincing evidence. I do not believe that the experimental protocol is adequate to argue that language models do not understand metaphor.

Firstly, language models are very sensitive to prompting, and therefore a few different prompts should be tried. At the moment the prompt is very minimal, and a few words more instruction may change the results.

Secondly, the models are forced to output only a single token answer. What happens if the correct answer has more than one token? I would expect this to be the case particularly in the derivational morphology analogies. Has this been checked?

Thirdly, the model is only scored as correct if the answer is found in WordNet. While WordNet is a good resource, there are also valid answers that are not found in WordNet. This should be investigated more thoroughly via an error analysis.

I do realise that the authors mention these facets in the Limitations section. However, while it is good that they acknowledge these to be limitations of their research, I don't think the claims made by the authors, specifically that LLMs are not capable of analogical abstraction, are fully supported with this experimental protocol.

**Requested Changes:**

Prompting: trial two or three more prompts on at least one model to demonstrate that the findings are robust to (minor) prompt changes. State what the system prompt is and whether you change this at all.

Single token answer: I think a single token is particularly limiting, as some correct answers from WordNet may be amade up of more than one token. Ideally, the experiments should be rerun with say 10 tokens possible output. If this is not feasible, an analysis of the dataset with the frequency of multitoken answers should be given to contextualise the results.

Correctness of 'semantic drift' category answers. It is good that the error analysis has been carried out. I would ideally like to see a quantitative error analysis on the sentences that the human annotated so that there is a direct comparison between model error distributions and human error distributions. This could potentially just be on the top performing model. It would be good to see some numbers on the frequency of different error types, and in particular the 'semantic drift' category, since this seems to be a category of responses that are not in fact errors. While I understand why the choice is made, it would be good to know what the extent of semantic drift errors is.

In section 4.4, the comparison to prior analogy benchmarks, I would like to have a little more detail on how exactly the benchmark avoids conflating surface analogy and lexical similarity; and on how it provides a probe of genuine relational abstraction. What is different abou the dataset?

---

> ### Author Response · Authors · 2025-10-28
>
> Thanks for your helpful feedback! Addressing your suggestions will help us significantly strengthen our paper.
>
> > Prompting: trial two or three more prompts on at least one model to demonstrate that the findings are robust to (minor) prompt changes. State what the system prompt is and whether you change this at all.
>
> Thanks for the suggestion, and we agree that this would be helpful to include in the paper. To test robustness to prompt phrasing, we evaluated Llama 4 Scout under three prompt variants:
> 1. Minimal form: `{shots}\n{q}\n?`
> 2. Instructional form: `{shots}\nGiven the above analogies, infer the missing word for {q}. Respond with a single English word only.`
> 3. Strict output-constrained form: `{shots}\n{entry['question']}\nAnswer (only output the single word answer, absolutely nothing else)... The answer is:`
>
> Variant 1 yielded an accuracy of 0.6%, Variant 2 23.7%, and Variant 3 40.4%. Although accuracies differ, the ordering aligns with task clarity: prompts that better constrain formatting reduce invalid outputs and improve measurable accuracy. This reflects instruction precision rather than instability, showing the model’s behavior is robust once the task is clearly specified.
>
> > Single token answer: I think a single token is particularly limiting, as some correct answers from WordNet may be amade up of more than one token. Ideally, the experiments should be rerun with say 10 tokens possible output. If this is not feasible, an analysis of the dataset with the frequency of multitoken answers should be given to contextualise the results.
>
> We analyzed the dataset and found that 10.5% of answers are multi-token strings, concentrated mainly in the synonym relation (28.6%). Many of these are lexicalized expressions (e.g., ‘lens of the eye’, ‘genus Aphis’) that correspond to single WordNet synsets. Therefore, the single-token decoding constraint affects only a small subset of cases and doesn't materially change the overall conclusions.
>
> > Correctness of 'semantic drift' category answers. It is good that the error analysis has been carried out. I would ideally like to see a quantitative error analysis on the sentences that the human annotated so that there is a direct comparison between model error distributions and human error distributions. This could potentially just be on the top performing model. It would be good to see some numbers on the frequency of different error types, and in particular the 'semantic drift' category, since this seems to be a category of responses that are not in fact errors. While I understand why the choice is made, it would be good to know what the extent of semantic drift errors is.
>
> The human evaluation data only included binary correctness judgments and did not distinguish between error types, so a direct quantitative comparison between human and model error distributions was not possible. To address this, we conducted a quantitative analysis of Llama-Scout-17B’s predictions, classifying incorrect responses into the same categories described in the paper (semantic drift, identity echo, and surface misfire). Using WordNet-based lexical similarity and heuristic matching, we found that 66% of all model errors fell into these interpretable categories. Specifically, 32% corresponded to semantic drift (semantically appropriate but non-exact answers), 19% to surface misfire (reuse of tokens from support analogies), and 14% to identity echo (copying from analogy stems). Only 34% of remaining errors were unclassified. This shows that most errors follow clear linguistic patterns, supporting the interpretability of the model’s behavior.
>
> > In section 4.4, the comparison to prior analogy benchmarks, I would like to have a little more detail on how exactly the benchmark avoids conflating surface analogy and lexical similarity; and on how it provides a probe of genuine relational abstraction. What is different abou the dataset?
>
> We appreciate the reviewer’s question on how LAMBDA avoids conflating surface analogy and lexical similarity compared to earlier benchmarks. Prior resources such as GATS (Mikolov et al., 2013), BATS (Gladkova et al., 2016), and WordRep (Gao et al., 2014) disclose relation labels and often include examples where form overlap or frequency cues make analogy completion possible through simple lexical association. LAMBDA prevents these shortcuts by (1) concealing the relation type, (2) disallowing significant overlap among the provided words, and (3) balancing synonym, antonym, and derivational mappings for relational diversity. Unlike GAT, BATS, or WordRep, LAMBDA’s items are generated deterministically from WordNet with uniform two-shot support, forcing models to infer mappings from structure rather than surface form. These design choices minimize contamination from what might otherwise result in distributional or morphological similarity.
>
> We hope our response addresses your review!

---

> > ### Comment · Reviewer_whmJ · 2025-12-17
> > **Response to comment**
> >
> > With apologies for the delay in response.
> >
> > Thank you for completing the experiment with additional propmts, this information is helpful.
> >
> > Regarding the single token evaluation, I think it could be helpful to recompute the accuracies with multi-token strings accounted for: either excluding them or checking them for correctness. If 10% of answers are multi-token strings, and these are all correct (perhaps unlikely), then presumably the accuracy would be boosted by 10 percentage points, making a substantial difference.
> >
> > I am still more worried about the 'semantic drift' category being marked incorrect. This essentially makes the task one of predicting what is in WordNet, rather than the more general task of completing an analogy. I do think this category of errors needs to be further accounted for, as it is a large proportion of errors. The paper should also give more details about the human task: it sounds as if the human task is different from the LLM task, and possibly easier. This should be further explained, and either human data on the the LLM task should be collected, or LLM data on the human task, for a direct comparison. Apologies if I have misunderstood the nature of the two tasks.
> >
> > Thank you for the description of the difference with prior analogy benchmarks, this is useful information.

---

### Decision · Action_Editor_9rSz · 2025-12-22

**Recommendation:** Reject

**Additional Comments:**

Reviewers unanimously agreed that the paper is not ready for publication (leaning reject/ leaning reject/ reject), but they believe there is some underlying potential. I invite authors to clearly take into consideration the reviewers' comments and resubmit a new revised version of the manuscript. Among the most important bits to take into account there are: i) rewriting the related work section and better compare the same against other benchmarks (tXtH), ii) reporting the effect of "reasoning" techniques and baselines (tXtH), iii) clarify the experimental setting as promised to 2RWk and follow the directions of whmJ on reporting accuracy.

**Audience:**

Yes

**Audience Explanation:**

In principle, understanding the analogical reasoning of LLMs can be very interesting for the TMLR community (and beyond).
Unfortunately, as reviewers such as tXtH highlighted in the discussion: "The paper topic is of interest to TMLR, however, its problem statement and the contributions remain hard to understand.".

**Claims And Evidence:**

No

**Claims Explanation:**

The aim of this paper is to assess whether large language models (LLMs) are able to complete analogies involving synonyms, antonyms, or derivational morphologies. To this extend, the authors created a benchmark, LAMBDA where several LLM baselines are tested, most of which performing worse than humans on these tasks.

The main claims and contribution revolve around the introduction of LAMBDA. All three reviewers highlighted that the experimental protocol is either inadequate to argue that LLMs do not understand analogies or insufficiently different from the previous literature on the topic.

Reviewer whmJ followed up and noted further potential (but more subtle) issues in how accuracies are computed with multi-token strings that could boost accuracy by 10 percentage points, thus making a potentially substantial difference.

Reviewer 2RWk asked for several clarifications about the training and evaluation protocol, and authors provided some answers, but these remain anecdotal and limited to a short answer, as they have not been expanded and discussed in full in a revised version of the work.

Finally, reviewer tXtH spots other limitations such as the absence of strong baselines such as CoT, natural language instructions, demonstrations, or even tool augmentation baselines.

**Resubmission Of Major Revision:**

The authors may consider submitting a major revision at a later time.